# CALINCA—A Novel Pipeline for the Identification of lncRNAs in Podocyte Disease

**DOI:** 10.3390/cells10030692

**Published:** 2021-03-20

**Authors:** Sweta Talyan, Samantha Filipów, Michael Ignarski, Magdalena Smieszek, He Chen, Lucas Kühne, Linus Butt, Heike Göbel, K. Johanna R. Hoyer-Allo, Felix C. Koehler, Janine Altmüller, Paul Brinkkötter, Bernhard Schermer, Thomas Benzing, Martin Kann, Roman-Ulrich Müller, Christoph Dieterich

**Affiliations:** 1German Center for Cardiovascular Research (DZHK), Partner Site Heidelberg/Mannheim, Im Neuenheimer Feld 669, 69120 Heidelberg, Germany; swttalyan@gmail.com; 2Section of Bioinformatics and Systems Cardiology, Klaus Tschira Institute for Integrative Computational Cardiology and Department of Internal Medicine III, Im Neuenheimer Feld 669, 69120 Heidelberg, Germany; Magdalena.Smieszek@uni-heidelberg.de; 3Department II of Internal Medicine and Center for Molecular Medicine, University of Cologne, Faculty of Medicine and University Hospital Cologne, 50931 Cologne, Germany; samantha.filipow@uk-koeln.de (S.F.); michael.ignarski@uk-koeln.de (M.I.); he.chen@uk-koeln.de (H.C.); lucas.kuehne@uk-koeln.de (L.K.); linus.butt@uk-koeln.de (L.B.); johanna.hoyer-allo@uk-koeln.de (K.J.R.H.-A.); felix.koehler@uk-koeln.de (F.C.K.); paul.brinkkoetter@uk-koeln.de (P.B.); bernhard.schermer@uk-koeln.de (B.S.); thomas.benzing@uk-koeln.de (T.B.); martin.kann@uk-koeln.de (M.K.); 4Cologne Excellence Cluster on Cellular Stress Responses in Aging-Associated Diseases (CECAD), University of Cologne, 50931 Cologne, Germany; 5Institute for Pathology, Diagnostic and Experimental Nephropathology Unit, University of Cologne, Faculty of Medicine and University Hospital Cologne, 50931 Cologne, Germany; heike.goebel@uk-koeln.de; 6Cologne Center for Genomics, University of Cologne, 50931 Cologne, Germany; jaltmuel@uni-koeln.de

**Keywords:** kidney, glomerulus, podocyte, focal-segmental glomerulosclerosis, FSGS, long non-coding RNA, lncRNA, RNAscope

## Abstract

Diseases of the renal filtration unit—the glomerulus—are the most common cause of chronic kidney disease. Podocytes are the pivotal cell type for the function of this filter and focal-segmental glomerulosclerosis (FSGS) is a classic example of a podocytopathy leading to proteinuria and glomerular scarring. Currently, no targeted treatment of FSGS is available. This lack of therapeutic strategies is explained by a limited understanding of the defects in podocyte cell biology leading to FSGS. To date, most studies in the field have focused on protein-coding genes and their gene products. However, more than 80% of all transcripts produced by mammalian cells are actually non-coding. Here, long non-coding RNAs (lncRNAs) are a relatively novel class of transcripts and have not been systematically studied in FSGS to date. The appropriate tools to facilitate lncRNA research for the renal scientific community are urgently required due to a row of challenges compared to classical analysis pipelines optimized for coding RNA expression analysis. Here, we present the bioinformatic pipeline CALINCA as a solution for this problem. CALINCA automatically analyzes datasets from murine FSGS models and quantifies both annotated and de novo assembled lncRNAs. In addition, the tool provides in-depth information on podocyte specificity of these lncRNAs, as well as evolutionary conservation and expression in human datasets making this pipeline a crucial basis to lncRNA studies in FSGS.

## 1. Introduction

Chronic kidney disease (CKD) affects almost 10% of the global population and is one of the most important independent risk factors for cardiovascular morbidity and mortality [1]. In humans, each kidney contains about 1 million nephrons made up of the actual filtration unit—the glomerulus—and the tubulus system, which determines the volume and composition of urine to be excreted. Glomerular disorders are the pre-dominant cause of kidney diseases leading to end-stage renal failure. The glomerulus consists of a capillary convolute containing a three-layered filtration barrier: Fenestrated endothelial cells are covered by a specialized basement membrane followed by podocytes, a post-mitotic epithelial cell type located on the urinary side of the barrier. Podocytes form primary and secondary foot processes and are crucial to the function of the filter [2]. Consequently, and due to their postmitotic nature as well as their limited capacity of regeneration, podocyte disorders play a central role in most—if not all—glomerular diseases. Cytoskeletal re-arrangements resulting in podocyte effacement are the common hallmark of podocyte injury and the loss of podocytes results in glomerular scarring, i.e., glomerulosclerosis [3]. Importantly, proteinuria—which is the direct clinical consequence of podocyte injury—is directly associated with disease progression in both cardiac and renal disease [4,5]. Focal-segmental glomerulosclerosis (FSGS) is a classic example of a podocytopathy. However, FSGS is rather a histopathological pattern of injury resulting from multiple pathogenic mechanisms than a uniform disease. Furthermore, secondary FSGS caused by systemic diseases such as arterial hypertension has to be clearly separated from the primary disease. Consequently, it is not surprising that clinical trials including all FSGS patients have failed to provide targeted treatment options [6]. At the moment, the treatment of FSGS is still unspecific and based on immunosuppression and blockade of the renin-angiotensin-aldosterone axis (RAAS). A better understanding of the molecular mechanisms underlying podocyte injury is urgently needed not only to design tailored therapeutic strategies but also to allow for a subclassification of FSGS by pathophysiology. Here, both external factors and podocyte-intrinsic defects are expected to play a role (reviewed by D’Agati et al. [7]). Regarding podocyte cell biology, the research was focused on proteins and protein-coding genes in the past. However, only 2% of all human transcripts possess a coding potential [8]. Interestingly, the non-protein coding component of the transcriptome shows greater tissue and context specific expression patterns than the coding genome and plays an important role in phenotypic variation between individuals and species [9]. This fact, on the one hand, clearly shows that the primary research focus on coding RNAs—followed by the majority of scientists over decades—comes at the risk of entirely missing a crucial aspect of cellular and molecular biology. On the other hand, considering the lack of knowledge, it is clear that studying non-coding RNAs (ncRNA) in FSGS bears a great potential to identify novel disease pathways. Much ncRNA research at the beginning of the 21st century focused on microRNAs (miRNAs)—with, e.g., highly interesting results regarding miR-193a in FSGS [10,11]. Meanwhile, various additional types of ncRNA molecules have been identified—including long non-coding circular RNAs—and have been implicated to play important roles in numerous cellular processes [12,13]. LncRNAs are defined as non-coding transcripts of more than 200 nucleotides and are transcribed by RNA polymerase 2. However, beyond this definition, the term lncRNAs refers to a heterogenous group of transcripts regarding their genomic organization and contains transcripts overlapping with other genes (both sense and antisense) as well as enhancer RNAs (eRNAs) and intergenic transcripts [14]. As a note of caution, when working with lncRNAs, it is important to keep in mind that some transcripts that are annotated as lncRNAs are in fact coding [15,16]—an aspect that needs to be considered and addressed both bioinformatically and experimentally. LncRNAs generally harbor a rather low sequence conservation, while recent evidence suggests that they show much higher functional, structural, and positional conservation across evolution [8,14,17]. Current estimates assume the existence of at least 20,000 lncRNAs in mammals ranging up to more than 200,000 in humans [18,19] (RNAcentral.org). Interestingly, a small number of lncRNAs had already been identified in the 80s and 90s, which—both as to their function, e.g., X-chromosome inactivation by XIST or inhibition of Igf2 by H19 and their nature—were regarded rare and exotic exceptions back then [20,21]. However, little to nothing is known regarding the function of the vast majority of these transcripts. The establishment of novel techniques in the field of RNA biology—both regarding computational biology and functional wet-lab experiments—coupled with an increasing amount of data over the last 5–10 years has greatly changed this situation. Efficient studies on the lncRNA function in biology and disease in a wide range of fields have thus become possible only recently. This work revealed a multitude of functions in cell biology including epigenetic regulation, transcriptional and posttranscriptional control, and adaptation of nuclear but also modulation of the stability/function of protein interaction partners [22]. Based on these findings, the importance of lncRNAs in the development and disease of a variety of organ systems has been underlined by the recent literature [23,24,25,26,27,28,29,30,31,32]. The vast majority of publications on lncRNAs in the kidney have remained merely descriptive and characterized the lncRNA expression in a large variety of cell culture and animal models [33]. Diabetic nephropathy (DN) is one of the few examples in which significant progress has been made. As a specific and highly interesting example, the lncRNA TUG1 was shown to be repressed upon high glucose exposure and to modulate mitochondrial bioenergetics in diabetic nephropathy by recruiting PGC-1α to its own promoter [34]. Regarding FSGS, published data on the involvement of lncRNAs still remain extremely scarce and have only been provided by two studies from the same group up to now. In both studies, the authors used transcriptome analyses in human tissue to identify upregulated lncRNAs in FSGS patients [35,36]. In general, the lack of a simple and readily available tool to identify candidate lncRNAs involved in the disease based on existing expression data has hampered systematic studies in this field. Here, we present CALINCA—a pipeline solving this problem in FSGS by providing tools to identify podocyte-enriched lncRNAs that are differentially regulated in FSGS models and conserved in evolution.

## 2. Materials and Methods

### 2.1. RNAscope

Kidneys from 12-week-old, three wildtype FVB/N mice were fixed in 4% formaldehyde, embedded in paraffin and 4.5 µm sections were cut and placed on Superfrost^®^ Plus glass slides (Thermo Scientific, Waltham, MA, USA). Samples were processed and stained using the commercially available in situ hybridization assay for FFPE samples—RNAscope 2.5 HD Assay—BROWN (cat no. 322310, Advanced Cell Diagnostics (ACD), Inc., Newark, CA, USA) and RNAscope probes for following lncRNAs targets: *Wt1os*, *4921504A21Rik*, *Gm10824*, and *XLOC_024349* (ACD, Inc, Newark, CA, USA). Representative images were captured using the Slide Scanner Leica SCN400 system (Leica Biosystems, Wetzlar, Germany) and prepared in the Aperio ImageScope 12.4.3 software (Leica Biosystems, Wetzlar, Germany). More details as well as the probes are provided in the Appendix A.

### 2.2. QPCR

For the detection of lncRNA expression in the glomeruli and the whole mouse kidneys, we performed quantitative RT-PCR analyses. Isolation of glomeruli and qPCR was carried out with magnetic beads, as described previously [37,38]. Briefly, total RNA was extracted using the Direct-zol RNA Kit (Zymo Research, Irvine, CA, USA). The cDNA was synthesized using the High-Capacity cDNA Reverse Transcription Kit (Applied Biosystems, Waltham, MA, USA). The qPCR was performed using custom TaqMan PrimeTime assays (Integrative DNA Technologies, Coralville, IA, USA) and the 7900HT Fast Real-Time PCR System (Applied Biosystems, Waltham, MA, USA). More details as well as primer sequences are contained in the Appendix A.

### 2.3. Animal Maintenance and Permissions

All animal experiments were conducted in accordance with European, national, and institutional guidelines and were permitted by the State Office of North Rhine-Westphalia, Department of Nature, Environment and Consumer Protection (LANUV NRW, Germany; animal approval AZ 81-02.04.2018.A325, AZ 2019.A085, and AZ 84-02_04_2014_A372). Experimental mice were kept in individually ventilated cages (Greenline GM500m Tecniplast, West Chester, PA, USA) at 22 °C and a humidity of 55% under a 12-h light cycle with unlimited access to water and food in a specific and pathogen free animal facility of the CECAD Research Center, University of Cologne, Germany.

### 2.4. Genetic Mouse Models

As described previously [2], podocinR231Q and podocinA286V mice were separately generated in our in vivo research facility (CECAD Research Center University of Cologne, Germany) using CRISPR-Cas9 based mutagenesis. Subsequently, podocinR231Q and podocinA286V mice were crossed to compound-heterozygosity. Genotyping was performed according to standard procedures using DNA isolated from ear biopsies. For podocinR231Q, DNA was visualized using gel electrophoresis following PCR-based amplification. For podocinA286V, DNA was amplified by PCR and analyzed by Sanger sequencing. The *Wt1* heterozygous deletion model has been described extensively in the literature [37]. All mice were kept in a pure C57Bl/6 background.

### 2.5. Adriamycin Treatment

Mice were obtained from Janvier Labs (Le Genest-Saint-Isle, France). Adriamycin nephropathy was induced in 11-week-old, male BALB/cJRj-wildtype mice via injection of Adriamycin at a concentration of 12 mg/kg body weight [39]. Male BALB/cJRj-wildtype mice of the same age served as control animals. For injections, the mice were anesthetized with isoflurane. Adriamycin, solubilized in 0.9% sodium chloride, was then administered intravenously via a tail vein cannula. After Adriamycin application, the mice were housed individually and examined daily for weight loss, tail vein necrosis, and abnormal behavior. The day of injection was counted as Day 0. Animals were euthanized on Day 5 and the tissue was collected for glomeruli isolation.

### 2.6. Preparation of Glomeruli and Isolation of Podocytes

Preparation of glomeruli and FACS-sorting of podocytes was performed as previously described [40,41]. Briefly, dynabeads M-450 (in Hank’s balanced salt solution, HBSS) were used to perfuse renal arteries after kidney dissection. Kidneys were minced and digested at 37 °C for 15 min (digestion solution: Collagenase II 300 U/mL (Worthington, Worthington, OH, USA), pronase E 1 mg/mL (Sigma-Aldrich, Darmstadt, Germany), and DNAse I 50 U/µL (Applichem, Darmstadt, Germany) in HBSS). The resulting suspension was sieved twice (100 µm) and glomeruli were separated using a magnetic particle concentrator. A glomerular single cell suspension was generated by incubation in a digestion solution at 37 °C for 40 min. GFP-positive cells were FACS-sorted on a BD FACSAria™ III cell sorter (Franklin Lakes, NJ, USA) (after sieving through a 40 µm filter).

Total RNA extraction was performed using the RNeasy RNA extraction kit (Qiagen, Germantown, MD, USA) according to the manufacturer’s protocol. RNA integrity was assessed using the Tape Station system (Agilent, Santa Clara, CA, USA), only samples reaching an RNA integrity number ≥8 were used for RNA-seq.

### 2.7. RNA-Sequencing

Wildtype whole kidney and FACS-sorted podocyte RNA-seq datasets were used as previously published by our groups [37,42]. Briefly, separate libraries were prepared both after polyA-RNA enrichment and ribosomal RNA depletion and sequenced on an HiSeq4000 sequencer (Illumina, San Diego, CA, USA) with PE75 read length. For the glomerular RNA-seq datasets, libraries were prepared with the TruSeq (Illumina, San Diego, CA, USA) stranded ribo zero gold protocol. After library validation and quantification (Agilent 4200 tape station), equimolar amounts of the library were pooled. Pools were quantified using the Peqlab KAPA Library Quantification Kit (Roche, Basel, Switzerland) and the 7900HT Sequence Detection System (Applied Biosystems, Waltham, MA, USA) and sequenced on an NovaSeq6000 sequencer (Illumina, San Diego, CA, USA) with PE100 read length. More details are available in the Appendix A.

### 2.8. Human Tissue

Human specimens were derived from healthy kidney tissue from a tumor nephrectomy after obtaining informed consent. All procedures were approved by the local institutional review board (#20-1206, Ethikkommission, Uniklinik Köln).

### 2.9. Data Analysis

#### 2.9.1. Read Processing and Mapping

Illumina RNA-seq reads were pre-processed with Flexbar 3 [43] for quality clipping and adapter removal. We used Bowtie2 [44] and mouse reference transcripts (rRNA, tRNA) to subtract t/rRNA reads in silico. All the remaining reads were aligned against the mouse genome using STAR (2.6.0c) [45], guided by the EnsEMBL v90 reference annotation.

#### 2.9.2. Transcript Assembly and Abundance Estimation

We used the following datasets to perform a transcriptome de novo assembly: FACS sorted Podocytes, Glomeruli wildtype samples, and whole kidney control samples. All of them were prepared with a ribo-zero library preparation strategy (see above). Our initial assembly was performed with Stringtie 1.3.5 [46] independently on each of the aforementioned tissue types. Subsequently, we merged the results and compared them against the Ensembl 90 reference using cufflinks 2.2.1 merge. Then, we used the resulting annotation file to estimate the RNA abundance with Stringtie 1.3.5 in the two podocyte-specific libraries (ribo-zero and polyA-selected RNA).

#### 2.9.3. Selection of Potential lncRNA Candidates

Our initial selection step across our transcriptome assembly was based on transcript length ≥200 bp and RNA expression cutoff ≥1 FPKM.

Then, we used TransDecoder 5.5 (https://github.com/TransDecoder) to identify candidate coding regions (ORF in sense orientation ≥50 aa in length) within all the selected transcript sequences generated. Non-coding sequences were classified based on a dynamic ORF length cutoff [47], TransDecoder log-likelihood score, and start codon prediction (position specific scoring matrix). The longest ORF was always selected, when candidates were contained one within the other.

#### 2.9.4. LncRNA Candidate Downstream Analysis

Sequence or gene order conservation with the human genome: We detected conserved gene orders (syntenic regions) based on the EnsEMBL protein coding gene annotation in man and mouse. Briefly, our Cyntenator software (latest version from November 2021) was used to compute all local gene order alignments based on gene coordinates and BLASTP scores [48]. NcRNAs adjacent to protein-coding genes were predicted as candidate orthologs. The nucleotide-level sequence conservation was detected by the alignment of lncRNA transcripts sequences from mouse against all human transcripts (n = 200, 310) annotated in reference GTF annotations (GRCh38.90.gtf). We only retained sequence alignments with an identity of more than 80% and alignment length of more than 100 bp. These criteria were used to assign additional ortholog pairs and are motivated by an approach for ultraconserved sequence element detection [1]. We assessed the expression of predicted candidate human orthologs using four non-tumour whole kidney GTEX samples. Protein Homology Search: All complete ORFs (i.e., with proper start and stop codons) which passed the dynamic ORF length cutoff were subjected to an additional BLASTX-based protein homology search using Swiss-Prot (cutoff: 50 bits).

Differential gene expression in FSGS disease models: We used edgeR (v3.24.3, [49]) to assess lncRNA gene locus expression changes. Briefly, we bundled all predicted lncRNA candidates into a new gene annotation set (excluding all protein-coding transcripts) to obtain read counts. Then, we used a two-factor design (age, condition) for the *WT1*-ko and *NPHS2*-ko model and a conventional one-factor design for the Adriamycin model. All loci with an FDR ≥ 0.05 in any comparison were retained as significant. Moreover, we performed an analysis of tissue specific expression using the tissue specificity index (TSI [50]). A TSI index score of ≤0.80 is used to predict the tissue specific enrichment.

#### 2.9.5. Re-Analysis of scRNA-seq Datasets

We obtained scRNA-seq control datasets from Kidney Glomeruli (Chung et al. [51]). Specifically, NCBI SRA accessions ctrl1: SRR11300654 and SRR11300655, ctrl2: SRR11300658 and SRR11300659, ctrl3: SRR11300660 and SRR11300661. We used the 10x Cell Ranger 3.0.2 software to process the data. We employed Seurat 3.2.2 [52] with default settings to generate UMAP plots and to identify podocytes based on cell-specific marker gene expression: *Wt1*, *Nphs1, Nphs2,* and *Mafb*. Only cells that express all four markers were labelled as podocytes. We tested for podocyte-specific gene expression with a Wilcoxon test using two groups: Podocytes vs. other cell types. All p-values are reported on calinca.dieterichlab.org and in Appendix A.

### 2.10. Transcriptome Data Availability

For the podocyte RNA-seq datasets, raw and processed sequencing data have been deposited in GEO (GSE64063) (accessible at http://www.ncbi.nlm.gov/geo/query/acc.cgi?acc=GSE64063). Regarding the wildtype kidney RNA-seq datasets, the RNA-seq primary data are available at https://www.ebi.ac.uk/arrayexpress/experiments/E-MTAB-7982. The glomerular RNA-seq data were uploaded to GEO (BioProject ID PRJNA715735).

## 3. Results

### 3.1. Bioinformatic Pipeline Design to Identify lncRNAs Involved in FSGS

In order to allow for the identification of lncRNA candidates involved in FSGS with a potential role in human disease, two aspects needed to be solved. Firstly, appropriate transcriptome datasets generated using comparable methodology were required. Secondly, the analysis workflow could not be based on the available tools for coding RNAs considering the specific challenges regarding lncRNAs—i.e., exclusion of coding potential, determination of evolutionary conservation, and identification of novel transcripts. The selection of sample types for RNA-seq was based on the following considerations. Dysregulation of lncRNAs in FSGS should not be based on a single model only and be determined before overt glomerular scarring. Consequently, we sequenced RNA obtained from glomeruli from three different mouse models. *Wt1* heterozygous deletion mutants (*Wt1^+/−^*) and *Nphs2* compound heterozygous mice (Pod^R231Q/A286V^) were analyzed as genetic models at two timepoints each (4 and 12 weeks of age). Both models reliably lead to proteinuria due to FSGS and have been characterized extensively by our group in the past [1,2]. Additionally, Adriamycin treatment was used as a pharmacological model to induce podocyte disease, as described in the literature [3]. To confirm the induction of podocyte disease, urinary albumin-to-creatinine ratios were measured on the fifth day after Adriamycin injection (Appendix A). To address lncRNA cell-type specificity whole kidneys, glomeruli and FACS-sorted podocytes from wildtype mice were analyzed in parallel (Figure 1, see also CALINCA flowchart on calinca.dieterichlab.org). To allow for optimal usage of these datasets, a bioinformatic pipeline (CALINCA) was set up addressing the key challenges associated with lncRNA transcript analyses (Figure 1). After read processing/mapping and transcript assembly for both annotated and novel lncRNAs, candidates are stringently examined for protein coding potential. Cell-type specificity is analyzed using the tissue specificity index (TSI) [50] and conservation in humans is determined based both on sequence and synteny. The CALINCA website (calinca.dieterichlab.org) provides interactive tools to create both user-defined tables and graphs to make all aspects of the pipeline accessible. Here, in addition to the points set forth in Figure 1, podocyte-specific expression can be interrogated using published scRNA-seq data and an insight on the expression of lncRNA candidates in humans is provided using GTEX kidney cortex RNA-seq data.

### 3.2. Characteristics of lncRNA Expression in the Kidney

An analysis of the datasets described revealed renal expression of 48,055 lncRNA transcripts after removal of transcripts containing ORFs above the defined length cutoff or showing an FPKM <1. About two thirds of these transcripts had previously been annotated, whilst one third comprises novel lncRNAs (Figure 2A, see also https://calinca.dieterichlab.org). In addition, 21,514 of these transcripts are conserved in human by sequence, 35,620 by synteny. In the context of conservation, we introduced a second step to remove putatively coding transcripts using a protein homology filter. As expected, this reduces the number of transcripts conserved by sequence (by about 32%) and by gene order conservation (by 19%). Podocytes express 20,942 lncRNA transcripts and these are derived from 13,199 genes (Figure 2A). We found 1,500 lncRNAs that show high podocyte-specific expression (TSI ≥ 0.8). The putative orthologues of 464 of these 1500 lncRNAs were found to be expressed in human kidney cortex data obtained from GTEX without de novo assembly, i.e., allowing only for a retrieval of annotated lncRNAs. A set of 879 podocyte-specific lncRNA transcripts remain after filtering for DNA sequence or gene order conservation and protein homology. Moreover, we computed the tissue specificity index for 15,155 novel transcripts, in which we had enough sequencing coverage. Our analysis indicated that expression of these novel transcripts is distributed across all of the three renal compartments analyzed with a tendency of the highest TSI in podocytes (39.7% podocytes, 34.9% glomeruli, 25.4% kidney (non-glomerular), Figure 2B). Figure 2C shows the overlap between the different transcript features highlighting a small set of 334 transcripts, which meet all listed criteria. Intriguingly, novel lncRNAs harbor more exons and tend to be longer than the annotated transcripts (Figure 2D).

### 3.3. Dysregulation of lncRNAs in FSGS Models

Since our main focus were lncRNAs dysregulated in podocytes we now focused on podocyte-expressed and conserved lncRNA gene loci in the three FSGS models. To this end, we established a lncRNA-only genome annotation and computed lncRNA gene loci abundance based on this annotation. A detailed list of dysregulation over all lncRNA transcripts is provided in Appendix A and can be visualized in a user-defined manner on the CALINCA website (https://calinca.dieterichlab.org). In addition, 379/2270/1833 out of 9789 tested lncRNA gene loci are differentially regulated in the Wt1^+/−^/ Pod^R231Q/A286V^/Adriamycin model (FDR < 0.05), respectively (Figure 3A, Appendix A). Eighty-nine lncRNA gene loci are significantly dysregulated in all three models. We further dissected this analysis to uncover coherently regulated lnc RNA gene loci (see bottom of Figure 3A). We see a high degree of coherent gene regulation between the Pod^R231Q/A286V^/Adriamycin and the Wt1^+/−^/ Pod^R231Q/A286V^ disease model. We integrated our findings from Figure 2D with the set of dysregulated candidates in Figure 3A. Two hundred and forty-one out of the 757 high-confident lncRNA loci overlap with the candidates contained in our three models (Figure 3B). In addition, we independently tested the cell-type specific expression in a published 3′ end scRNA-seq dataset (Chung 2020) (Figure 3C). We could detect the expression of 203 lncRNA gene loci (out of the 241 candidates) strongly supporting our prediction of podocyte-specific expression for the majority of lncRNAs based on bulk RNA-seq (ribo zero) data. Figure 3D highlights this finding for two known loci with high TSI values for podocyte-specificity (Wt1os and 4921504A21Rik) in UMAP projections. An examination of podocyte-specific expression of all lncRNAs is provided on https://calinca.dieterichlab.org.

### 3.4. Experimental Validation of FSGS lncRNA Candidates

To confirm the validity of the results obtained by CALINCA, we used three independent approaches for a subset of the lncRNAs found to be enriched in podocytes, conserved in human and dysregulated in at least one of our three FSGS models. Firstly, all of these candidates were examined in a published dataset containing transcriptomes of FACS-sorted glomerular cells and comparing podocyte to all non-podocyte cells (Boerries et al.) [2]. Fifty-five percent of the 1500 podocyte-enriched lncRNA transcripts identified by our TSI analysis could be retrieved as podocyte-specific in these data, as well (information on this analysis for each specific transcript is provided on https://calinca.dieterichlab.org). Secondly, we selected six lncRNAs for further confirmation of enrichment in glomeruli compared to whole-kidney RNA samples. Three out of six (Wt1os, 4921504A21Rik, XLOC_024349) could be confirmed as podocyte-specific in both the datasets of FASC-sorted glomerular cells from Boerries et al. and the scRNA-seq dataset from Chung et al. [3]. Two of the remaining three (Gm26759, Gm10824) were confirmed by at least one of these additional analyses and only Gm28876 was not found as podocyte-enriched in either of the two (see https://calinca.dieterichlab.org). Using qPCR, five out of six lncRNAs were significantly enriched in glomeruli, in which Gm10824 showed a trend towards glomerular enrichment (Figure 4A, Appendix A). Thirdly, visualization of lncRNA expression using in situ hybridization appeared an important step to validate both expression and cell-type specificity. We chose RNAscope technology [53] for this purpose due to the importance of sensitivity and specificity considering the low expression of most lncRNAs compared to coding genes. RNAscope stainings were performed for four of the lncRNA candidates. For three of the lncRNAs most strongly enriched in glomeruli, as shown by qPCR (Wt1os, 4921504A21Rik, XLOC_024349), RNAscope signal is indeed limited to glomeruli (Figure 4B, Appendix A). Importantly, XLOC_024349, a novel lncRNA identified by CALINCA is one of these candidates showing validity of the de novo assembly (Appendix A). In line with the qPCR results, expression of Gm10824 was detected in tubular epithelial cells, even though at lower levels compared to glomeruli (Appendix A). Importantly, the human orthologue of *Wt1os-*W*T1-AS—*is also specifically expressed in glomerular cells when examined by RNAscope in human tissue (Figure 4C, Appendix A).

## 4. Discussion

The development of novel treatment strategies for FSGS as well as other glomerular diseases has been hampered by a lack of knowledge regarding causative factors and molecular mechanisms underlying podocyte pathophysiology. This shortcoming also limits the possibility to classify FSGS by pathobiology, an extremely important goal in a situation in which many different causes lead to the same histopathological picture. Since lncRNAs have not been studied systematically in FSGS to date, this field bears an extraordinary potential for the identification of novel pathomechanisms and therapeutic targets. However, lncRNA research comes with a couple of challenges—especially in the context of human disease. This includes bona-fide definition of truly non-coding transcripts and de novo detection of previously not annotated lncRNAs. Even more importantly—especially in translational research—evolutionary conservation has to be addressed and relies on criteria that differ much from coding genes since sequence conservation is often limited. Consequently, the available tools cannot easily be transferred from coding RNAs to ncRNAs. Additionally, when comparing disease to healthy states as well as different cell types, transcriptome data generation needs to be homogenous regarding tissue preparation and sequencing technology.

CALINCA provides a solution for these challenges and integrates a multi-step bioinformatic pipeline with a row of RNA-seq datasets in FSGS models and healthy kidneys including FACS-sorted podocytes. However, the focus on this specific disease and tissue type is primarily guided by the RNA-seq data used as input—in general, the code underlying CALINCA can easily be adapted to other disease scenarios in different tissues.

Our work now provides the first comprehensive atlas of lncRNAs that fulfil several important aspects to be considered key candidates in FSGS. These candidates are specific to or enriched in podocytes, the key cell type in disorders of the glomerular filter. In addition, the lack of coding potential is ensured by several steps in the algorithm and dysregulation in FSGS is examined in several mouse models. CALINCA also identifies novel lncRNAs. Importantly, the algorithm does not only confirm conservation by sequence and synteny but also considers human expression data to ensure that these genes are indeed transcribed in humans. This is a crucial point, since—obviously—research in disease models should be limited to conserved transcripts, an aspect associated with much uncertainty regarding lncRNAs in the past. Here, we confirm the podocyte-specific expression of one of these lncRNAs—WT1-AS—in human tissue using RNAscope [53]. This approach is very helpful to actually show cell-type specific expression and provides single molecule sensitivity, highly specific staining of the target RNA, and improved detection of degraded RNA. Importantly, RNAscope can be automated and used for RNA-protein co-expression analyses making it a potential entry point towards the use of lncRNA quantification in human diagnostics [54].

Making large-scale RNA-seq datasets available to the scientific community in a fashion that allows for simple and efficient interrogation is an extremely important aspect to make full use of the power of such data. This is especially true when implementing a novel pipeline for new RNA biotypes such as lncRNAs. The CALINCA website (calinca.dieterichlab.org) provides such a tool allowing for both global and transcript-specific analysis of our datasets. Importantly, the generation of user-defined tables is accompanied by the possibility to generate graphs for the visualization of the selected analyses. Using this option shows that dysregulated lncRNAs contain a much higher proportion of podocyte-specific transcripts in models using podocyte-specific interventions (such as *Wt1^+/−^* and *Pod^R231Q/A286V^*) compared to Adriamycin toxicity, which directly affects all cells of the glomerulus (calinca.dieterichlab.org, histograms examples 1-4, and boxplots examples 1–3). To provide another example, CALINCA can also be used to examine glomerular lncRNAs previously described in glomerular disease. As mentioned above, most data are available in the context of diabetic nephropathy. Consequently, we checked CALINCA regarding four examples in the context of diabetic nephropathy. *Tug1*, an important player in the metabolic response of podocytes in diabetes mellitus [34], is indeed also expressed in podocytes in our data and—in line with current knowledge—conserved in humans. This is also the case for *Malat1*, a mediator of cellular damage in diabetic glomerulopathy [55]. However, in addition, *Malat1* is dysregulated in the *Pod^R231Q/A286V^* model (up at 4 weeks, down at 12 weeks) and may consequently play a role in podocyte biology beyond diabetes mellitus. *Neat1* mediates diabetic damage through glomerular activation of Akt/mTOR signaling. CALINCA shows *Neat1* to be significantly enriched in podocytes. Interestingly, all three FSGS models lead to a dysregulation of *Neat1* suggesting that this lncRNA may be a central player in podocyte pathobiology. Furthermore, knockdown of *Pvt1* has been implicated in diabetic nephropathy as a mediator of podocyte apoptosis [56]. However, in our datasets, *Pvt1* is not expressed in podocytes pointing towards the fact that podocyte expression of this lncRNA may actually be exclusive to the setting of high glucose. Taken together, these examples show how CALINCA can be used to extend our knowledge on lncRNAs in glomerular disease and to design future studies in this field.

Of course, our current study does not address the lncRNA function in disease yet. Here, further work—primarily using knockout mouse models—will be required.

## 5. Conclusions

Taken together, CALINCA examines compartment-specific expression of conserved lncRNAs in the healthy kidney, and will be a beneficial tool to study lncRNA involvement in renal (patho-) physiology in general. By including data from several FSGS models, CALINCA is a powerful tool to identify conserved lncRNAs in FSGS. Using this novel pipeline will now facilitate lncRNA research in FSGS but also other glomerular and tubular renal diseases. To allow for dynamic user-friendly access, the analyses are provided through an interactive website (https://calinca.dieterichlab.org).

## Figures and Tables

**Figure 1 cells-10-00692-f001:**
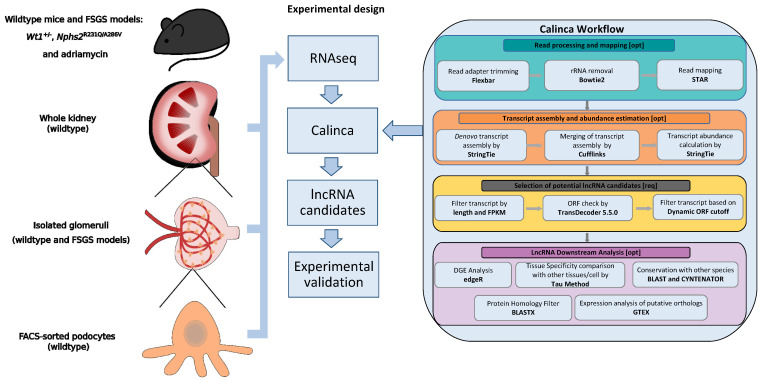
The CALINCA pipeline. Experimental design and stepwise visualization of the CALINCA pipeline developed for this study. RNA-seq data generated from podocytes, glomeruli, and whole kidneys of wildtype mice alongside glomeruli from three focal-segmental glomerulosclerosis (FSGS) models (Wt1^+/−^, Nphs2^R231Q/A286V^, Adriamycin) were analyzed according to the depicted CALINCA workflow. For a more detailed overview of the datasets, see “CALINCA Flowchart” on https://calinca.dieterichlab.org. Briefly, long non-coding RNA (lncRNA) expression in renal compartments is quantified after read processing and mapping. In addition, the quantification of annotated lncRNAs novel transcripts is detected using a reference-guided de novo assembly. The lncRNA candidates were defined based on open reading fram (ORF) cutoffs. These candidates are then checked for tissue-specificity using the tissue specificity index (TSI) method and evolutionary conservation based on synteny and sequence.

**Figure 2 cells-10-00692-f002:**
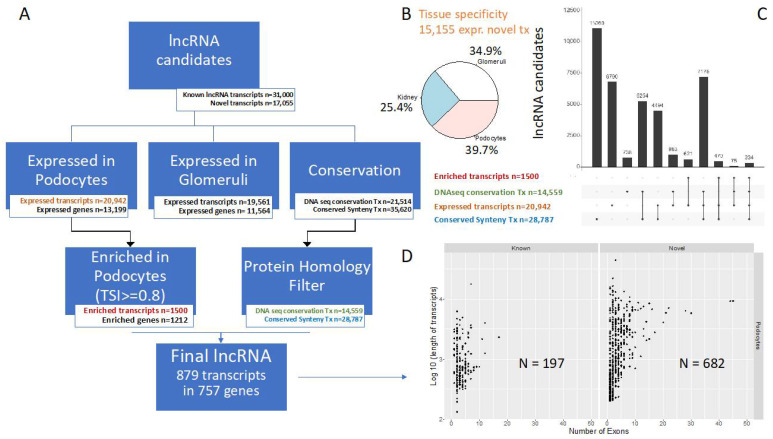
CALINCA identifies 879 conserved podocyte lncRNAs. (**A**) Our workflow to define the set of podocyte-specific lncRNAs (see Methods). A final set of 879 conserved and podocyte-specific lncRNAs is defined. (**B**) Tissue specificity of novel podocyte-expressed lncRNAs candidates. We identified 15,155 out of 20,942 expressed transcripts as novel (i.e., not represented in the reference annotation). The tissue specificity is defined by the maximal TSI value. (**C**) Stratification of lncRNA candidates by expression and conservation (see the colored tags in panel A). The overlap between different lncRNA candidate properties is shown as set intersections. (**D**) Transcript length and number of exons stratified by annotation status (known/novel) for the final set of 879 conserved and podocyte-enriched lncRNA transcripts.

**Figure 3 cells-10-00692-f003:**
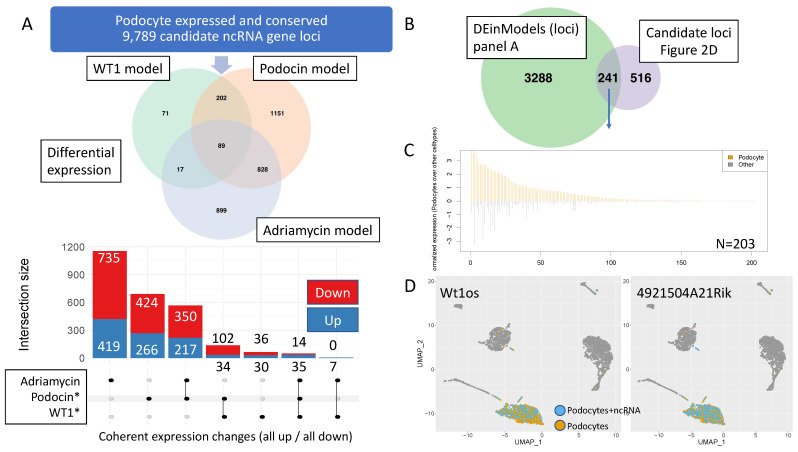
Dysregulation of lncRNAs in three mouse models of FSGS and validation of cell-type specificity using scRNA-seq data. (**A**) Venn diagram of differential lncRNA candidate gene loci expression across three disease models (top). Additional details on co-regulation across disease models is given at the bottom bar chart. *The two-factor model (age, condition) is used for Podocin and WT1. (**B**) Overlap of differential gene expression with the 757 conserved, podocyte-specific lncRNA candidate loci from Figure 2D. (**C**) Re-assessment of differentially regulated candidate loci from Figure 2D with regards to podocyte-specificity using single cell data from Chung et al. JASN 2020. We could identify 203 out of 241 lncRNA loci as expressed in 3′ scRNA data with the majority being podocyte-specific. (**D**) Uniform Manifold Approximation and Projection graphs (UMAPs) of two examples of highly podocyte-specific and conserved lincRNAs in scRNA-seq data (Chung et al. JASN 2020): Wt1os and 4921504A21Rik.

**Figure 4 cells-10-00692-f004:**
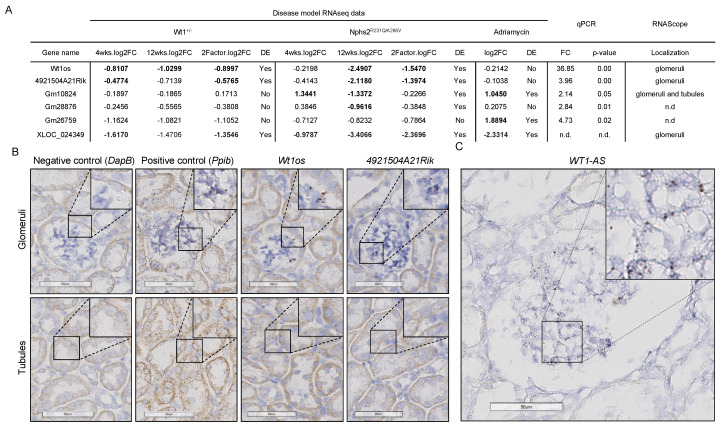
The qPCR and RNAscope validate podocyte-specific lncRNAs as defined by CALINCA. A) Table showing six lncRNAs dysregulated in at least one of the FSGS models, the glomerular expression which was validated by qPCR and/or RNAscope. Regarding Wt1^+/−^ and Pod^R231Q/A286V^ models, a transcript is classified as differentially expressed (DE/Yes) if it is significantly regulated (adjusted *p*-value < 0.05) in at least one of the time points (4 weeks, 12 weeks) or in the two-factor model. The same cutoffs apply for the Adriamycin model with a one-time point only. For the full table, refer to Appendix A or calinca.dieterichlab.org. (**A**) Visualization of the qPCR data is provided in Appendix A. (**B**) Representative images of glomeruli and tubules analyzed with custom designed RNAscope probes for lncRNAs Wt1os and 4921504A21Rik. Both lncRNAs are specifically detected in glomerular cells only, whilst the positive control shows a signal in both glomeruli and tubuli. Additional images as well as the results for Gm10824 and XLOC_024349 are provided in Appendix A. Target lncRNAs and controls were detected with the RNAscope 2.5 HD—brown assay on FFPE mouse kidney tissue sections. Probe binding is visualized as punctate brown dots. Counterstain: Hematoxylin (blue). Scale bar: 60 µm. (**C**) The human ortholog of Wt1os (WT1-AS) is expressed in glomerular cells. Representative image of a human glomerulus analyzed with a custom designed RNAscope probe for WT1-AS. The lncRNA was detected with the RNAscope 2.5 HD—brown assay on a formalin-fixed, paraffin-embedded (FFPE) human kidney tissue section. Probe binding is visualized as punctate brown dots. Counterstain: Hematoxylin (blue). Scale bar: 90 µm. More images as well as the images showing lack of expression in tubuli are provided in Appendix A.

## Data Availability

RNAseq data are available as specified in Section 2.10 *Transcriptome Data Availability* and the analyses are provided on https://calinca.dieterichlab.org.

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
