# Peer review of "CALINCA—A Novel Pipeline for the Identification of lncRNAs in Podocyte Disease"

_cells, 2021, doi:10.3390/cells10030692_

Round 1
Reviewer 1 Report
The author presented a bioinformatic pipeline CALINCA to identify novel lncRNAs through three different models. The authors validated several lncRNAs through staining. 1. Is CALINCA very specific for lncRNA? Could it be used to other ncRNAs, such as circle RNA? 2. Before RNA sequencing, do the author check phenotype of those models? The data should be shown in the figures to make us believe that the models are good. 3. Besides the Glomerulus specific scRNA-seq, do the author map the candidates in other whole kidney sc RNA-seq, such as Dr. BD Humphreys or Dr. Susztak? 4. When validating the lncRNA, the author only performed qpcr in isolated glomeruli. The isolated tubule should be added as a control to compare the expression with the one in the glomeruli. 5. Do the author have zoom out images of Figure 4C? The current picture shows tubule has weak signaling.Author Response
- Is CALINCA very specific for lncRNA? Could it be used to other ncRNAs, such as circle RNA?
Yes, CALINCA is specific for lncRNAs at the moment. We are happy to include additional ncRNA species in future versions. To this end, we will use circtools, a software that was implemented in the Dieterich Lab. However, since circRNAs come at different computational challenges compared to lncRNAs, this would go beyond the scope of the manuscript at hand.
- Before RNA sequencing, do the author check phenotype of those models? The data should be shown in the figures to make us believe that the models are good.
We thank the reviewer for raising this question. The adriamycin and Wt1 heterozygous deletion mouse models used in our study are commonly used murine models of FSGS. Adriamycin injection-induced nephropathy is a widely used pharmacological disease model, showing podocyte injury followed by glomerulosclerosis, which resembles human FSGS. The phenotype was described previously by a number of publications, e.g. Menke et al., 2003[1] as well as work by members of our own group (Kann et al. 2015)[2]. The PodR231Q/A286V model was recently established by members of our group and induces the reliable development of FSGS (Butt et al. 2020)[3]. Since pharmacological models are usually more variable than genetic models, we added albumin-to-creatinine ratio measurements as confirmation of podocyte for the adriamycin mice sequenced in our study. Besides, we added these arguments in the main body of the manuscript in section 3.1.
- Besides the Glomerulus specific scRNA-seq, do the author map the candidates in other whole kidney sc RNA-seq, such as Dr. BD Humphreys or Dr. Susztak?
We have considered published whole kidney single cell sequencing data (e.g. Park et al. Science 2018). However, none of the published experiments reaches a sufficient sequencing depth.
It turns out that the sequencing depth per cell is an order of magnitude lower than for the newer glomerular scRNAseq data that were used in CALINCA (Chung et al. 2018). To provide an example, we could not identify any cell that expresses all four standard podocyte markers simulatenously ("Wt1","Nphs1","Nphs2",“Mafb“). Considering the generally much lower expression of lncRNAs than coding RNAs as well as the very low proportion of podocytes in whole kidney scRNAseq data, this analysis would not have added to the current data. Upon availability of suitable datasets in the future this may change. However, for now, we believe that the qPCR and RNAScope proven glomerular enrichment of our candidates over whole kidney paired with the podocyte-specificity in glomerular single-cell RNAseq sufficiently shows enrichment in podocytes.
- When validating the lncRNA, the author only performed qpcr in isolated glomeruli. The isolated tubule should be added as a control to compare the expression with the one in the glomeruli.
This is an important point. We addressed this concern by not only analyzing glomeruli but comparing these to qPCR data from whole kidneys (made up primarily by tubular cells, whilst podocytes only account for less than 1%). These data allow for the conclusion that the lncRNAs examined are indeed significantly enriched in glomeruli. The fold changes - glomeruli over whole kidney - are given in figure 4A and supplementary figure 2.
- Do the author have zoom out images of Figure 4C? The current picture shows tubule has weak signaling.
We thank the reviewer for this suggestion. We believe that adding another zoom-in panel to figure 4C would make it very crowded. Instead we have added zoom-in panels to supplementary figure 3 and 4.. Zooming in the tubular region shows that the RNAscope signal for WT1-AS in the tubular regions is much weaker and not distributed in the punctate pattern as expected and shown for both the positive control and the lncRNAs in glomeruli. The tubular staining pattern mimicks the signal observed for the negative control. Consequently, we are convinced that this signal is unspecific background and believe that this is now nicely shown by the new suppl. Figures 3 and 4.
Reviewer 2 Report
Talyan et al have presented the study of lncRNAs analysed using CALINCA method, as a potential tool detecting a podocyte pathology in FSGS and other glomerular injury in mice model. Presented data are interesting and worth to implement in further studies diagnosing and monitoring different types of kidneys disease especially in humans. I would suggest to proceed the manuscript for publication and present the study results for nephrologist scientific community.
Author Response
Talyan et al have presented the study of lncRNAs analysed using CALINCA method, as a potential tool detecting a podocyte pathology in FSGS and other glomerular injury in mice model. Presented data are interesting and worth to implement in further studies diagnosing and monitoring different types of kidneys disease especially in humans. I would suggest to proceed the manuscript for publication and present the study results for nephrologist scientific community.
We would like to thank the reviewer for reviewing our manuscript and are glad about this positive feedback.
Round 2
Reviewer 1 Report
My concerns have been addressed by the authors.